# The Efficacy of Rotary, Reciprocating, and Combined Non-Surgical Endodontic Retreatment Techniques in Removing a Carrier-Based Root Canal Filling Material from Straight Root Canal Systems: A Micro-Computed Tomography Analysis

**DOI:** 10.3390/jcm9061989

**Published:** 2020-06-25

**Authors:** Tarek-Fahed Alakabani, Vicente Faus-Llácer, Ignacio Faus-Matoses, Celia Ruiz-Sánchez, Álvaro Zubizarreta-Macho, Salvatore Sauro, Vicente Faus-Matoses

**Affiliations:** 1Department of Stomatology, Faculty of Medicine and Dentistry, University of Valencia, 46010 Valencia, Spain; tarekalakabani@hotmail.com (T.-F.A.); fausvj@uv.es (V.F.-L.); ignacio.faus@uv.es (I.F.-M.); ceruizsan@gmail.com (C.R.-S.); vicente.faus@uv.es (V.F.-M.); 2Department of Endodontics, Faculty of Health Sciences, Alfonso X El Sabio University, 28691 Madrid, Spain; 3Department of Dentistry, Faculty of Health Sciences, CEU Cardenal Herrera University, 46115 Valencia, Spain; salvatore.sauro@uchceu.es

**Keywords:** micro-computed tomography, non-surgical endodontic retreatment, reciprocating movement, root canal filling material

## Abstract

The aim of this study is to analyze and compare the efficacy of three non-surgical endodontic retreatment techniques in removing a carrier-based root canal filling material from straight root canal systems. The study was performed on 99 single-rooted extracted teeth using the ProTaper Gold endodontic rotary system up to the F2 file (Dentsply Maillefer, Baillagues, Switzerland), which were sealed with GuttaCore (Dentsply Maillefer, Ballaigues, Switzerland) and AH plus epoxy resin sealer (Dentsply DeTrey, Konstanz, Germany) and randomly assigned to the following non-surgical retreatment techniques: ProTaper Retreatment endodontic rotary instruments (D1–D3 files, Dentsply Maillefer, Ballaigues, Switzerland; *n* = 33, PTR), Reciproc Blue endodontic reciprocating instrument (R50, VDW, Munich, Germany; *n* = 33, RCB50), and a combined root canal retreatment technique between Gates-Glidden drills (sizes #3 and #2, Dentsply Maillefer, Ballaigues, Switzerland) and Hedstrom files (file size 35, 30, and 25, Dentsply Maillefer, Ballaigues, Switzerland; *n* = 33; H-GG). All of the teeth were submitted twice to a micro-computed tomography (micro-CT) scan, before and after non-surgical endodontic retreatment procedures. The volume of root canal filling material (mm^3^), volume of remaining root canal filling material (mm^3^), non-surgical endodontic retreatment working time (min), proportion of remaining root canal filling material (%), and efficacy of root canal filling material removal between the non-surgical endodontic retreatment techniques were analyzed using ANOVA one-way statistical analysis. Statistically significant differences were observed between the proportions of remaining root canal filling material of PTR and H-GG (*p =* 0.018), between the non-surgical endodontic retreatment working times (min; *p* < 0.001), and between the efficacies of root canal filling material removal by the non-surgical endodontic retreatment techniques (*p* = 0.009). However, the non-surgical endodontic retreatment systems allow for similar carrier-based root canal filling material removal.

## 1. Introduction

Bacterial infection plays an important role in establishing pulp tissue inflammation, which may lead to subsequent pulp necrosis and the formation of periapical lesions [1]. The complete removal or at least a significant reduction of the bacterial load during root canal treatment is an important factor determining the final prognosis of the root canal treatment. However, the development of apical periodontitis has been reported in 44.9% of studied cases [2], mainly related to persistent or secondary endodontic infections [3]. Non-surgical endodontic retreatment is recommended after unsuccessful root canal treatment [4]; however, the prognosis of non-surgical endodontic retreatment is often associated with an insufficient disinfection of the root canal system, inadequate obturation, missed root canals, under-extended or over-extended root canal filling material, or coronal microleakage [5,6,7,8]. The relevance of completely removing the root canal filling material to achieve a fully disinfected root canal system has been highlighted [9], because incomplete root canal filling material removal prevents dentinal tubule disinfection [10]. In most cases, the etiology of endodontic failure is related to persistent or secondary endodontic infections [3]. Antibacterial irrigation solutions such as sodium hypochlorite (NaOCl) can penetrate up to 130 µm into dentinal tubules, while some bacterial species are able to penetrate more than 250 μm deep and adhere to the collagen present in human serum, leaving bacteria harboring in deeper layers, accessory canals, anastomoses, and fins [11]. Secondary infections are often linked to facultative anaerobic Gram-positive microorganisms, particularly *Enterococcus faecalis,* which has been shown to be highly resistant to conventional antimicrobial agents and is able to invade dentinal tubules, causing reinfection in the root canal system [12,13].

Several non-surgical endodontic retreatment techniques have been proposed in order to fully enhance the removal of the root canal filling material, including the use of hand files, ultrasonic tips, and endodontic rotary and reciprocating instruments [14,15]. Many endodontic instruments have been used in non-surgical endodontic retreatment; however, none has reported a complete removal of the root canal filling material from the root canal system. In addition, the marginal sealing capability of carrier-based root canal filling materials and techniques may influence the removal capability of the endodontic rotary and reciprocating instruments [16].

Many measurement procedures have been used to assess the volume of the root canal filling material removal, including tooth splitting, diaphanization, conventional periapical radiography, or digital imaging [17]. However, tooth splitting is an invasive procedure that may spread the root canal filling material, and conventional periapical radiography and digital imaging produce two-dimensional images of the three-dimensional root canal system [18]. Micro-computed tomography (micro-CT) scans offer a high-resolution non-invasive technique, which provides accurate three-dimensional digital files that may be repeated as required [19]. Therefore, a micro-CT imaging technique has been recommended for endodontic research, especially in order to analyze the remaining root canal filling material [20].

The aim of this study is to analyze and compare the efficacy of three non-surgical endodontic retreatment techniques for removing carrier-based root canal filling material from straight root canal systems, with a null hypothesis (H0) stating that there would be no difference between the non-surgical endodontic retreatment techniques with regard to the removal of carrier-based root canal filling material from the straight root canal systems.

## 2. Experimental Section

### 2.1. Study Design

Ninety-nine single-rooted anterior teeth (lower central incisors) extracted for periodontal reasons, with curvatures of <10°, according to Schneider’s method [21], such as mature roots, the absence of root filling materials, calcium metamorphosis, and root resorptions, were selected in this study from the Department of Stomatology of the University of Valencia (Valencia, Spain), between January and March 2019. Sample calculation was performed using the ANOVA one-way test. A randomized controlled experimental trial was conducted in accordance with the principles defined in the German Ethics Committee′s statement for the use of organic tissues in medical research (Zentrale Ethikkommission, 2003), and was approved by the University of Valencia Ethics Committee (process no. H1512122849636). All of the patients gave their informed consent to transfer the teeth for the study.

### 2.2. Experimental Procedure

The teeth were digitally radiographed in a buccolingual and mesiodistal direction so as to standardize the samples. The crowns of the teeth were removed using a diamond disk under copious water cooling in order to obtain a standardized root length of 17 mm for all of the teeth. A size 8 K-file (Dentsply Maillefer, Baillagues, Switzerland) was inserted into the root canal system, until it was visible at the apical foramen under the operative microscope at 10× magnification (Zeiss Dental Microscope, Oberkochen, Germany).

The root canal systems were performed with a crown-down technique using the ProTaper Gold endodontic rotary system (F2, Dentsply Maillefer, Baillagues, Switzerland), using a 6:1 reduction handpiece (X-Smart plus, Dentsply Maillefer, Baillagues, Switzerland) and a torque-controlled motor with continuous rotation at 300 rpm and 2 N/cm torque, according to the manufacturer’s recommendations. The root canal systems were irrigated with 5 mL of 5.25% sodium hypochlorite (NaOCl; Clorox, Oakland, CA, USA) during the endodontic rotary instruments sequence. The final irrigation was performed with 5 mL of 5.25% NaOCl, 5 mL of 17% ethylenediaminetetraacetic acid (EDTA) (SmearClear, SybronEndo, CA, USA), 5 mL of 5.25% NaOCl, and 5 mL of sterile saline solution (Braun^®^, Melsungen, Germany) using an endodontic needle (Miraject Endo Luer, Hager and Werken, Duisburg, Germany) with a diameter of 0.3 mm inserted 1 mm into the working length. The contact between the irrigating solution and the surface of the root canal walls was improved by using a sonic device (Endoactivator^®^, Dentsply Sirona^®^, Ballaigues, Switzerland). Afterwards, the root canal system was dried with sterile paper points (Dentsply Maillefer, Ballaigues, Switzerland), and finally, each root canal system was sealed using an epoxy-amine resin-based sealer (AH Plus, Dentsply DeTrey, Konstanz, Germany) and a warm gutta-percha carrier-based system (GuttaCore, Dentsply Maillefer, Ballaigues, Switzerland) by heating a size 25 GuttaCore obturator in the ThermaPrep heater obturator oven (Dentsply Maillefer, Ballaigues, Switzerland) and subsequently introducing it into the root canal system at the working length. The GuttaCore obturator was cut at the cementoenamel junction and compacted following the manufacturer′s recommendations. Digital radiographs of the teeth were taken to ensure the outcome of the root canal filling procedure; whenever any voids in the obturation were observed, the specimen was discarded and replaced. Finally, the endodontic access cavity was temporarily sealed with Cavit (3M ESPE, Saint Paul, MN, USA), and the teeth were then stored in an incubator (mco-18aic, Sanyo, Moriguchi, Osaka, Japan) for 1 week (37 °C, 100% relative humidity).

Thirty days after the root canal treatments, the teeth were randomly distributed (Epidat 4.1, Galicia, Spain) into the following study groups: Group A, ProTaper Retreatment (PTR) endodontic rotary instruments (D1–D3 files, Dentsply Maillefer, Ballaigues, Switzerland; *n* = 33); Group B, Reciproc Blue (RCB50) endodontic reciprocating instrument (R50, VDW, Munich, Germany; *n* = 33); and Group C, combined root canal retreatment technique between Gates-Glidden drills (sizes #3 and #2, Dentsply Maillefer, Ballaigues, Switzerland) and Hedstrom files (file size 35, 30, and 25, Dentsply Maillefer, Ballaigues, Switzerland; *n* = 33; H-GG).

PTR was used to remove the carrier-based root canal filling material. The coronal third of the carrier-based root canal filling material was removed using the D1 file (30.09; 16 mm), while the middle and apical thirds were removed using the D2 file (25.08; 18 mm) and D3 file (20.07; 22 mm), respectively, using a crown-down technique with a 6:1 reduction handpiece and a torque-controlled motor with continuous rotation at 500 rpm and 4 N/cm torque, according to the manufacturer’s recommendations. Once the working length was reached, a brushing motion was applied to the root canal walls.

The carrier-based root canal filling material of the samples randomly assigned to the RCB50 study group (R50, VDW, Munich) was removed by a 6:1 reduction handpiece (Silver Reciproc; VDW, Munich, Germany) and a torque-controlled motor with continuous rotation at 300 rpm and 2 N/cm torque, according to the manufacturer’s instructions. RCB50 performed an in-and-out pecking motion with 3 mm amplitude. The instrument was removed from the canal and cleaned after three pecking motions until the RCB50 reached the working length.

The coronal and middle thirds of the carrier-based root canal filling material of the samples randomly assigned to H-GG were removed using sizes #3 and #2 Gates-Glidden drills. Furthermore, the apical third of the carrier-based root canal filling material was removed using Hedstrom files. The instruments made a circumferential quarter-turn, push-pull motion until the file size of 25 reached the working length.

The root canal systems were also irrigated with 5 mL of 5.25% NaOCl during the endodontic rotary instrument sequence. The final irrigation was performed with 5 mL of 5.25% NaOCl, 5 mL of 17% EDTA, 5 mL of 5.25% NaOCl, and 5 mL of sterile saline solution using an endodontic needle with a diameter of 0.3 mm inserted up to 1 mm into the working length. The contact between the irrigating solution and the surface of the root canal walls was improved by using a sonic device. All of the endodontic procedures were performed by a single clinician.

### 2.3. Micro-CT Scanning Procedures and Evaluation

All of the samples were submitted to a micro-CT scan twice (Micro-CAT II, Siemens Preclinical Solutions, Knoxville, TN, USA), regardless the study group, with the following exposure parameters: 80 kV, 500 mA, isotropic resolution of 21 μm, and 360° rotation. The first micro-CT scan (Micro-CAT II, Siemens Preclinical Solutions, Knoxville, TN, USA) was performed after the root canal treatment was performed in the PTR (Figure 1A), RCB50 (Figure 1C), and H-GG (Figure 1D) study groups, and the second micro-CT scan (Micro-CAT II, Siemens Preclinical Solutions, Knoxville, TN, USA) was performed after the non-surgical endodontic retreatment procedures in the PTR (Figure 1B), RCB50 (Figure 1D), and H-GG (Figure 1F) study groups.

### 2.4. Measurement Procedure

The micro-CT images were automatically reconstructed using Cobra software v.7 (Exxim Computing Corporation, Livermore, CA, USA) and were rendered with Amira 3D software v.6.0 for the remaining root canal filling material analysis (Thermo Scientific, Agawam, MA, USA). The micro-CT generated standard tessellation language (STL) digital files by means of a cloud of points that created a tessella network, representing three-dimensional objects as polygons composed of tessellas of equilateral triangles. The STL digital files obtained were imported to the FIJI software v.1.52 (National Institutes of Health, Bethesda, MD, USA), and a tooth alignment procedure was conducted by superimposing the STL digital files of each tooth before and after the non-surgical endodontic retreatment using the external root surface of the tooth as reference, using the best fit algorithm. Afterwards, a volumetric analysis of the carrier-based root canal filling material (mm^3^), a volumetric analysis of the carrier-based root canal filling material removed volume (mm^3^), a non-surgical endodontic retreatment working time (min), and the proportion of remaining carrier-based root canal filling material (%) were calculated by a blinded examiner using Equation (1):[B/A] × 100,(1)
where B is the volume of the root canal system after non-surgical endodontic retreatment (mm³) and A is the volume of the root canal system sealed by the root canal filling material (mm³)

In addition, the efficacy of the carrier-based root canal filling material removal was also analyzed. The carrier-based root canal filling material removal was determined to be successful if more than 95% of the carrier-based root canal filling material was removed. The root canal filling volumes were expressed in cubic millimeters, while the volume of the remaining carrier-based root canal filling material was expressed as a percentage.

### 2.5. Statistical Tests

All of the variables of interest were recorded for statistical analysis with SPSS 22.00 (Microsoft inc, Redmond, WA, USA) for Windows. The descriptive statistical analysis was expressed as mean and standard deviation (SD) for the quantitative variables. A comparative analysis was performed by comparing the mean volumetric analysis of the carrier-based root canal filling material (mm^3^), volumetric analysis of the carrier-based root canal filling material removed volume (mm^3^), non-surgical endodontic retreatment working time (min), proportion of remaining carrier-based root canal filling material (%), and the efficacy of carrier-based root canal filling material removal between the non-surgical endodontic retreatment techniques using an ANOVA one-way test; *p* < 0.05 was considered statistically significant.

## 3. Results

The mean and SD values for the carrier-based root canal filling material volume (mm^3^) are displayed in Table 1 and Figure 2.

The teeth randomly assigned to the H-GG study group showed slightly more root canal filling material volume (12.02 ± 3.99 mm^3^), followed by the PTR (11.63 ± 2.47 mm^3^) and RCB50 samples (10.70 ± 2.63 mm^3^). However, no statistically significant differences were observed (*p* > 0.05) between the carrier-based root canal filling material volumes of the H-GG, RCB50, and PTR study groups.

The mean and SD values for the carrier-based root canal filling material removed volume (mm^3^) are displayed in Table 2 and Figure 3.

The teeth randomly assigned to the H-GG study group showed slightly more carrier-based root canal filling material removal volume (11.56 ± 3.72 mm^3^), followed by the PTR (11.47 ± 2.45 mm^3^) and RCB50 samples (10.49 ± 2.65 mm^3^). However, no statistically significant differences were observed (*p* > 0.05) between the root canal filling material removal volumes of the H-GG, RCB50, and PTR study groups.

The mean and SD values for the non-surgical endodontic retreatment working time (min) are displayed in Table 3 and Figure 4.

Statistically significant differences were observed (*p* < 0.05) between the non-surgical endodontic retreatment working time of the non-surgical endodontic retreatment techniques. The teeth randomly assigned to the RCB50 study group showed a significantly lower non-surgical endodontic retreatment working time than the teeth randomly assigned to the PTR (*p* < 0.05) and H-GG (*p* < 0.05) study groups. In addition, statistically significant differences were also observed between the non-surgical endodontic retreatment working time of the PTR and H-GG study groups (*p* < 0.05).

The mean and SD values for the proportion of remaining carrier-based root canal filling material (%) are displayed in Table 4 and Figure 5.

Statistically significant differences were observed (*p* > 0.05) between the proportion of remaining carrier-based root canal filling material of the non-surgical endodontic retreatment techniques. The teeth randomly assigned to the PTR study group showed a significantly lower proportion of remaining carrier-based root canal filling material than the teeth randomly assigned to the H-GG study group (*p* < 0.05). However, no statistically significant differences were observed between the proportion of remaining carrier-based root canal filling materials of the RCB50 and PTR (*p* > 0.05) study groups and between the RCB50 and H-GG study groups (*p* > 0.05).

The PTR study group showed the highest rate of carrier-based root canal filling material removal efficacy, because 51.5% of the teeth randomly assigned to this non-surgical endodontic retreatment technique showed a more than 95% removal of carrier-based root canal filling material. However, only 21.2% of the teeth randomly assigned to the RCB50 and H-GG study groups showed a more than 95% removal of the root canal filling material. Statistically significant differences were observed (*p* < 0.05) between the carrier-based root canal filling material removal efficacy of the PTR and H-GG non-surgical endodontic retreatment techniques.

## 4. Discussion

The results obtained in the present study accept the null hypothesis (H0), which states that there would be no difference between the non-surgical endodontic retreatment techniques with regard to the removal of carrier-based root canal filling material from the straight root canal systems.

The results obtained in the present study show the combined root canal retreatment technique between Gates-Glidden drills and Hedstrom files removed a significantly higher proportion of remaining carrier-based root canal filling material from straight root canal systems than the ProTaper Retreatment endodontic rotary instruments (*p =* 0.018). In addition, the root canal retreatment technique combining Gates-Glidden drills and Hedstrom files used a significantly higher (*p* < 0.001) non-surgical endodontic retreatment working time (7.13 ± 0.87 min) than the ProTaper Retreatment endodontic rotary instruments (5.27 ± 0.72 min) and Reciproc Blue endodontic reciprocating instrument (2.93 ± 0.61 min). However, the carrier-based root canal filling material removal of the volume of the H-GG (11.56 ± 3.72 mm^3^), PTR (11.47 ± 2.45 mm^3^), and RCB50 samples (10.49 ± 2.65 mm^3^) did not show statistically significant differences (*p* = 0.283). This may be due to the time spent exchanging instruments during the instrumentation sequence. ProTaper Retreatment is a multiple-file endodontic rotary system constituting three endodontic instruments, Reciproc Blue is a single file endodontic reciprocating instrument, and the combined root canal retreatment technique between Gates-Glidden drills and Hedstrom files employs six instruments. In addition, the mean carrier-based root canal filling material volume of the combined root canal retreatment technique between the Gates-Glidden drills and Hedstrom files study group was slightly higher (12.02 ± 3.99 mm^3^) than the mean carrier-based root canal filling material volume of the ProTaper Retreatment endodontic rotary instruments study group (11.63 ± 2.47 mm^3^) and the mean carrier-based root canal filling material volume of the Reciproc Blue endodontic reciprocating instrument (10.70 ± 2.63 mm^3^); so there is more root canal filling material to remove and it needs more working time to achieve the desired results. Novel endodontic instruments have been used in non-surgical endodontic retreatment, and ProTaper Gold endodontic rotary instruments and Reciproc Blue endodontic reciprocating instruments have demonstrated their effectiveness in root canal filling removal [22]; however, none has reported a complete root canal filling material removal from the root canal system. Bago et al. reported a higher capability of the Reciproc endodontic reciprocating system to remove the root canal filling material from the root canal system than Reciproc Blue endodontic reciprocating system and ProTaper Retreatment endodontic rotary system [23]; however, none of these articles analyzed the efficacy of the endodontic rotary or reciprocating systems to remove a carrier-based root canal filling material. The GuttaCore carrier-based root canal filling material has demonstrated a high marginal sealing capability, especially in oval-shaped root canal systems; furthermore, it can penetrate to a depth of 96 µm and 48 µm into the dentinal tubules, located at 5 mm and 2 mm of the working length, respectively. The dentinal tubules tag removal produced by the GuttaCore carrier-based root canal filling material is a challenge during non-surgical endodontic retreatment, because preventing the complete disinfection of the root canal system and may lead to persistent or secondary endodontic infections [3].

This study analyzed and compared three non-surgical endodontic retreatment techniques: endodontic rotary instruments, endodontic reciprocating instruments, and combined root canal retreatment technique between Gates-Glidden drills and Hedstrom files. Various studies have reported that endodontic rotary [24,25] or reciprocating instruments [26,27] remove the root canal filling material more effectively, whereas other studies have reported that hand files are more efficient [28,29]. In the present study, straight root canals with curvatures of <10° were selected, standardizing the non-surgical endodontic retreatment procedures for all of the specimens. According to the results of the baseline scans, no statistically significant differences were found in the mean carrier-based root canal filling material volumes between the study groups before the non-surgical endodontic retreatment procedures, allowing for a reliable intergroup comparison. The micro-CT imaging technique was used to measure the carrier-based root canal filling material volume and the proportion of the carrier-based root canal filling material, as the technique provided a non-destructive, accurate, three-dimensional means of quantifying the carrier-based root canal filling material volume and the proportion of the carrier-based root canal filling material before and after non-surgical endodontic retreatment [30]. Cone beam computed tomography (CBCT) scans were also used to analyze the proportion of the remaining carrier-based root canal filling material into the root canal system, because it is easy and quick to perform, and is a repeatable and non-invasive technique; however, micro-CT scans allow more accurate images than CBCT scans [31]. The results showed that none of the non-surgical endodontic retreatment techniques removed the root canal filling material completely. This finding agrees with the outcomes of previous studies that have tested different instruments and techniques [6,7,15,16].

The present results also showed that the mean percentage of the remaining root canal filling material was significantly less with the ProTaper Retreatment endodontic rotary instruments study group than the combined root canal retreatment technique of the Gates-Glidden drills and Hedstrom files study group. This finding agrees with previous studies [24,32], probably because ProTaper Retreatment endodontic rotary instruments have a larger internal core and area, and a convex triangular cross section, variable taper, and a continuously changing helical pitch, a design that leads to effective cutting and coronal extrusion of the gutta-percha from the canal [14,33]. However, the comparison between the rotary and reciprocating endodontic techniques did not show statistically significant differences in the carrier-based root canal filling material removal, which is consistent with the results of previous studies [6,7]. This can be explained by the fact that the Reciproc Blue endodontic reciprocating system generates a longer counter-clockwise movement but shorter clockwise motion, maintaining the file in a more centered position in the canal [34,35]. This, in combination with the files′ tapered shape, produces a larger contact area between file and the carrier-based root canal filling material, making the removal action of the carrier-based root canal filling material as effective as the continuous rotation movement [33]. Nevertheless, the volume of the root canal filling material removed was higher after using the ProTaper Retreatment endodontic rotary instruments study group than the Reciproc Blue endodontic reciprocating system; this may be attributed to the brushing motion produced by the endodontic rotary instruments, the higher working time used, and the higher sequence of instruments used by the ProTaper Retreatment endodontic rotary instruments study group.

## 5. Conclusions

In conclusion, within the limitations of this study, the results show that the ProTaper Retreatment endodontic rotary instruments left a smaller proportion of remaining carrier-based root canal filling material than Reciproc Blue, and a combined root canal retreatment technique between Gates-Glidden drills and Hedstrom files. However, the non-surgical endodontic retreatment systems allow for a similar carrier-based removal of the root canal filling material, and none of them were able to remove carrier-based root canal filling material from a straight root canal system completely.

## Figures and Tables

**Figure 1 jcm-09-01989-f001:**
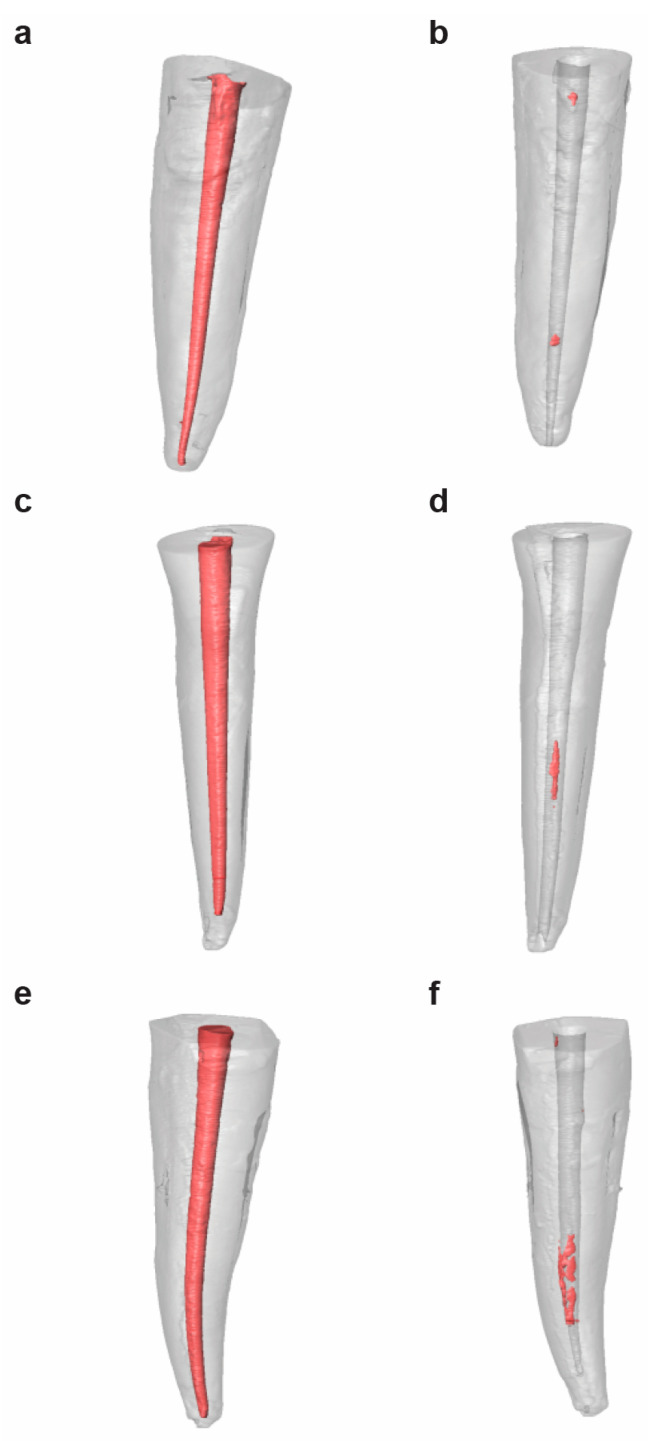
Reconstructed three-dimensional micro-computed tomography (micro-CT) images before (**A**,**C**,**E**) and after (**B**,**D**,**F**) non-surgical endodontic retreatment, according to the non-surgical endodontic retreatment techniques.

**Figure 2 jcm-09-01989-f002:**
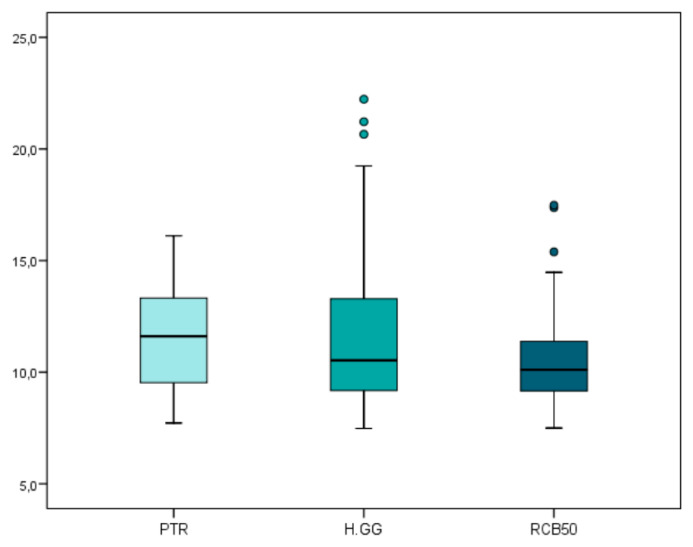
Box plot of the carrier-based root canal filling material volume (mm^3^) regarding the non-surgical endodontic retreatment techniques. PTR: ProTaper Retreatment endodontic rotary instruments; RCB50: Reciproc Blue endodontic reciprocating instrument; H-GG: combined root canal retreatment technique between Gates-Glidden drills and Hedstrom files.

**Figure 3 jcm-09-01989-f003:**
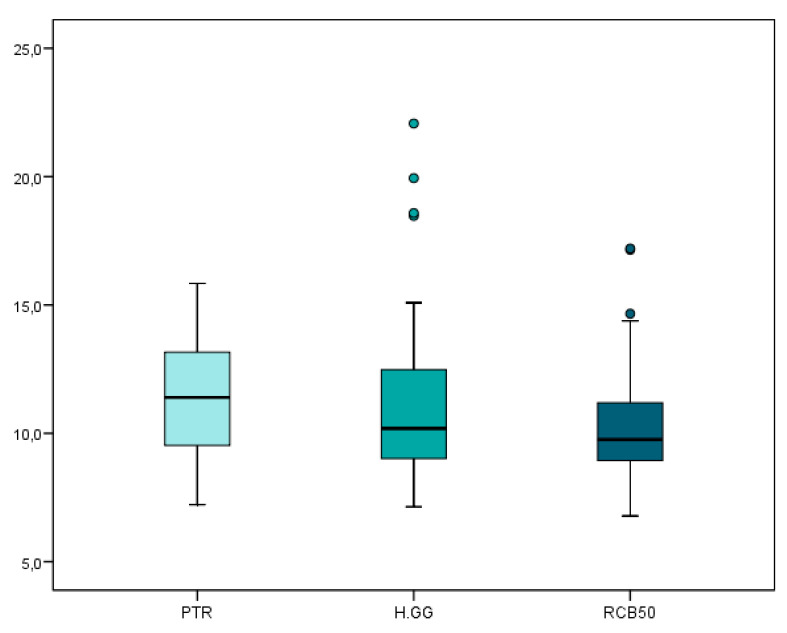
Box plot of the carrier-based root canal filling material removed volume (mm^3^) regarding the non-surgical endodontic retreatment techniques. PTR: ProTaper Retreatment endodontic rotary instruments; RCB50: Reciproc Blue endodontic reciprocating instrument; H-GG: combined root canal retreatment technique between Gates-Glidden drills and Hedstrom files.

**Figure 4 jcm-09-01989-f004:**
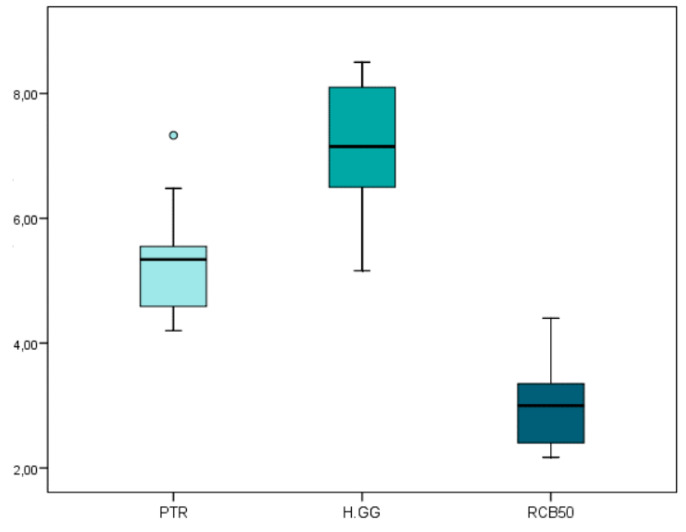
Box plot of the non-surgical endodontic retreatment working time (min) regarding the non-surgical endodontic retreatment techniques. PTR: ProTaper Retreatment endodontic rotary instruments; RCB50: Reciproc Blue endodontic reciprocating instrument; H-GG: combined root canal retreatment technique between Gates-Glidden drills and Hedstrom files.

**Figure 5 jcm-09-01989-f005:**
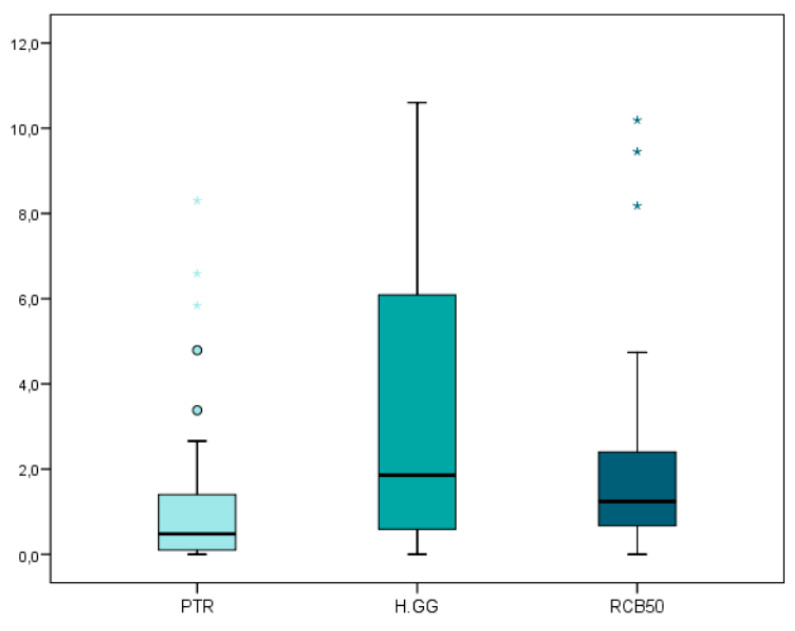
Box plot of the proportion of remaining carrier-based root canal filling material (%) regarding the non-surgical endodontic retreatment techniques. PTR: ProTaper Retreatment endodontic rotary instruments; RCB50: Reciproc Blue endodontic reciprocating instrument; H-GG: combined root canal retreatment technique between Gates-Glidden drills and Hedstrom files.

**Table 1 jcm-09-01989-t001:** Descriptive analysis of carrier-based root canal filling material volume (mm^3^), regarding the non-surgical endodontic retreatment study groups.

		*n*	Mean	SD	Minimum	Maximum
Root Canal Filling Material Volume	PTR	33	11.63 *	2.47	7.72	16.11
	RCB50	33	10.70 *	2.63	7.50	17.48
	H-GG	33	12.02 *	3.99	7.48	22.23

* Statistically significant differences between groups (*p* < 0.05); PTR: ProTaper Retreatment endodontic rotary instruments; RCB50: Reciproc Blue endodontic reciprocating instrument; H-GG: combined root canal retreatment technique between Gates-Glidden drills and Hedstrom files.

**Table 2 jcm-09-01989-t002:** Descriptive analysis of carrier-based root canal filling material removed volume (mm^3^) regarding the non-surgical endodontic retreatment study groups.

		*n*	Mean	SD	Minimum	Maximum
Root Canal Filling Material Remaining Volume	PTR	33	11.47 *	2.45	7.22	15.84
	RCB50	33	10.49 *	2.65	6.78	17.20
	H-GG	33	11.56 *	3.72	7.14	22.07

* Statistically significant differences between groups (*p* < 0.05). PTR: ProTaper Retreatment endodontic rotary instruments; RCB50: Reciproc Blue endodontic reciprocating instrument; H-GG: combined root canal retreatment technique between Gates-Glidden drills and Hedstrom files.

**Table 3 jcm-09-01989-t003:** Descriptive analysis of non-surgical endodontic retreatment working time (min) regarding the non-surgical endodontic retreatment study groups.

		*n*	Mean	SD	Minimum	Maximum
Non-Surgical Endodontic Retreatment Working Time	PTR	33	5.27 *	0.72	4.20	7.33
	RCB50	33	2.93 *	0.61	2.17	4.40
	H-GG	33	7.13 *	0.87	5.16	8.50

* Statistically significant differences between groups (*p* < 0.05). PTR: ProTaper Retreatment endodontic rotary instruments; RCB50: Reciproc Blue endodontic reciprocating instrument; H-GG: combined root canal retreatment technique between Gates-Glidden drills and Hedstrom files.

**Table 4 jcm-09-01989-t004:** Descriptive analysis of the proportion of remaining carrier-based root canal filling material (%) regarding the non-surgical endodontic retreatment study groups.

		*n*	Mean	SD	Minimum	Maximum
Remaining Root Canal Filling Material	PTR	33	1.43 *	2.09	0.00	8.30
	RCB50	33	2.07 *	2.57	0.00	10.19
	H-GG	33	3.52 *	3.66	0.00	10.60

* Statistically significant differences between groups (*p* < 0.05). PTR: ProTaper Retreatment endodontic rotary instruments; RCB50: Reciproc Blue endodontic reciprocating instrument; H-GG: combined root canal retreatment technique between Gates-Glidden drills and Hedstrom files.

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
