# Peer review of "The Efficacy of Rotary, Reciprocating, and Combined Non-Surgical Endodontic Retreatment Techniques in Removing a Carrier-Based Root Canal Filling Material from Straight Root Canal Systems: A Micro-Computed Tomography Analysis"

_jcm, 2020, doi:10.3390/jcm9061989_

Round 1

Reviewer 1 Report

Well prepared and meticulously deigned study, however,

there are some innate drawbacks in this experiment.

1) Size of the file in each Group is different that leads to the difference in the efficiency of removing the content in the root canal system which might negate the comparison per se.

2) Not only the nature (characteristics) of rotary file system, the rotation speed is different from each other.

This implicates that the comparison between the file itself is not at the same level of judgement.

3) The carrier-based gutta percha filling material contains two different materials in one core. Therefore, removal efficiency should be described separately because each file system works differently to 2 other materials. Direct comparison of the removal technique according to the remnant inside of the canal is clinically irrelevant and even misleading    

Author Response

Dear reviewer 1,

I’m pleased to resubmit the manuscript of the work entitled, “The Efficacy of Rotary, Reciprocating, and Combined Non-surgical Endodontic Retreatment Techniques in Removing a Carrier-Based Root Canal Filling Material from Straight Root Canal Systems: A Micro-Computed Tomography Analysis”.

Reviewer 1: English language and style are fine/minor spell check required.

Response: In order to adapt to the reviewer 1 comment, we have sent the manuscript to the English editing service of MDPI editorial (We attached the English Editing Certificate).

Reviewer 1: Size of the file in each Group is different that leads to the difference in the efficiency of removing the content in the root canal system which might negate the comparison per se. Not only the nature (characteristics) of rotary file system, the rotation speed is different from each other. This implicates that the comparison between the file itself is not at the same level of judgement.

Response: In order to respond to the reviewer 1 comment, we clarify that we are aware that the systems selected for the study are different in terms of size, design, rotation speed and type of movement. The objective of the study was to analyze and compare the efficacy of three non-surgical retreatment systems, following the manufacturers' recommendations. Other authors have also analyzed and compared non-surgical retreatment systems of different nature (characteristics): Alakabani TF, Faus-Llácer V, Faus-Matoses V. Evaluation of the time required to perform three retreatment techniques with dental microscope and ultrasonic activation for removing filling material from the oval root canal. J Clin Exp Dent. 2018; 10 (8): e810 ‐ e814, Zuolo AS, Mello JE Jr, Cunha RS, Zuolo ML, Bueno CE. Efficacy of reciprocating and rotary techniques for removing filling material during root canal retreatment. Int Endod J. 2013; 46 (10): 947‐953, Kanaparthy A, Kanaparthy R. The Comparative Efficacy of Different Files in The Removal of Different Sealers in Simulated Root Canal Retreatment- An In-vitro Study. J Clin Diagn Res. 2016; 10 (5): ZC130 ‐ ZC133, Joseph M, Ahlawat J, Malhotra A, Rao M, Sharma A, Talwar S. In vitro evaluation of efficacy of different rotary instrument systems for gutta percha removal during root retreatment channel. J Clin Exp Dent. 2016; 8 (4): e355 ‐ e360. Regardless, the reviewer's comments are appreciated and will be used for future studies where the influence of cross section, apical diameter, rotation speed, type of movement, etc. can be analyzed and compared.

Reviewer 1: The carrier-based gutta percha filling material contains two different materials in one core. Therefore, removal efficiency should be described separately because each file system works differently to 2 other materials. Direct comparison of the removal technique according to the remnant inside of the canal is clinically irrelevant and even misleading.

Response: In order to respond to the reviewer 1 comment, we clarify that unlike other rigid carrier-based systems, such as Thermafil, GuttaCore system is composed by a cross-linked gutta-percha carrier covered by gutta-percha. The internal core that performs the carrier functions is modified gutta-percha, with special properties that prevent it from melting during the heating process. However, this carrier based root canal filling material system has reported shorter non-surgical retreatment working times than those observed by other carrier based root canal filling material system such as Thermafil (Fracchia DE, Amaroli A, Angelis N, et al. GuttaCore Pink, Thermafil and Warm Vertically compacted gutta-percha retreatment: Time required and quantitative evaluation by using ProTaper files. Dent Mater J. 2020;39(2):229‐235). Regardless, the aim of this study was to analyze and compare the efficacy of three non-surgical endodontic retreatment techniques to remove a carrier-based root canal filling material from straight root canal systems; however we appreciate the reviewer's comment as it will lead to future studies.

We take this opportunity to thank the recommendations and suggestions made by the reviewer to improve the document.

Yours sincerely,

Reviewer 2 Report

First of all, English language needs to be highly revised. Major spelling so as grammatical errors can be found. Please consult a native English expert.

Other comments:

Several sentences have no meaning in English, need to be reformed.

The methods section is not totally understandable. Please describe more detailed or more clearly.

It is also not clear, why you have different volumes for the different groups, if the preparation method and sequence was exactly the same...

I also do not understand, why you explain the removed volume, why not the residual volume of the filling material? 

Author Response

Dear reviewer 2,

I’m pleased to resubmit the manuscript of the work entitled, “The Efficacy of Rotary, Reciprocating, and Combined Non-surgical Endodontic Retreatment Techniques in Removing a Carrier-Based Root Canal Filling Material from Straight Root Canal Systems: A Micro-Computed Tomography Analysis”.

Reviewer 2: Extensive editing of English language and style required.

Response: In order to adapt to the reviewer 2 comments, we have sent the manuscript to the English editing service of MDPI editorial (We attached the English Editing Certificate).

Reviewer 2: Several sentences have no meaning in English, need to be reformed.

Response: In order to adapt to the reviewer 2 comments, we have sent the manuscript to the English editing service of MDPI editorial (We attached the English Editing Certificate).

Reviewer 2: The methods section is not totally understandable. Please describe more detailed or more clearly.

Response: In order to respond to the reviewer 2 comments, we have sent the manuscript to the English editing service of MDPI editorial (We attached the English Editing Certificate), to make the methods more understandable.

Reviewer 2: It is also not clear, why you have different volumes for the different groups, if the preparation method and sequence was exactly the same...

Response: In order to respond to the reviewer 2 comments, we clarify that despite selecting the sample following the selection criteria and performing all the root canal systems with the same sequence and instruments in order to standardize the sample, the anatomy of the root canal systems of the selected teeth are slightly but not statistically significant different from each other. In summary, even though the authors did their best to standardize the dimensions of the root canal systems, the anatomical differences in the root canal systems of the selected teeth prevented them from being exactly the same. It is considered a limitation of the ex vivo study.

Reviewer 2: I also do not understand, why you explain the removed volume, why not the residual volume of the filling material?

Response: In order to respond to the reviewer 2 comments, we clarify that the removed volume is a parameter which depends only to the capacity of the endodontic rotary instrument to remove gutta-percha from the root canal system. However, the volume of residual gutta-percha not only depends on the capacity of the endodontic rotary instrument, but also on the entire volume of gutta-percha of the root canal system, and not all root canal systems has the same volume. That is, if one duct is very conical and another narrower, and we submitt them to non-surgical retreatment with the same instrument, both may have removed the same volume, but there will be a different residual volume of the root canal filling material. However, we have also analyzed the remaining carrier-based root canal filling material (%).

We take this opportunity to thank the recommendations and suggestions made by the reviewer to improve the document.

Yours sincerely,

Reviewer 3 Report

The paper is on the efficacy of guttapercha carrier based removal during retreatment performed by different endodontic instruments. Since it’s clear and comprehensive, some minor revisions have to be made to the manuscript before acceptance.

Abstract

The following sentence is too long and it should be reformulated due to make the paragraph more clear: I suggest: “All teeth were submitted twice to a micro- computed tomography (micro-CT) scan before and after non-surgical endodontic retreatment procedures. Volumetric analysis of the root canal filling material (mm3), volumetric analysis of the remaining root canal filling material (mm3), non-surgical endodontic retreatment working time (min), proportion of remaining root canal filling material (%) and the efficacy of root canal filling material removing between the non-surgical endodontic retreatment techniques were analyzed using ANOVA one-way statistical analysis.”

Introduction

Line 51: [5,6,7,8] should be written [5-8].

Materials and methods

Line 117: the sentence “…by introducing a size 25 GuttaCore obturator in the ThermaPrep heater obturator oven (Dentsply Maillefer, Ballaigues, Switzerland) to the working length.” Seems to be mispelled. Maybe the authors wanted to say that they have introduced the instrument before in the oven, then in the tooth to the working length. Please verify.

Once you’ve cited the manufacturer’s details of a product, it’s not necessary to repeat it after everytime you write the name of an instrument or whatever. Please, remove them all, instead of the first one.

Once you’ve citated the complete name of a product, it’s not necessary to re-write down it completely everytime you cite it: when and if it is possible, please use abbreviations like PTR, H-GG, etc.

Results

Please, provide only (p<0.05) and (p>0.05) instead of (p=x.xxx), when reporting statistical significance.

Line 268: “Statistically significant differences were 268 observed (p=0.009) between the carrier-based root canal filling material removal efficacy of the non-269 surgical endodontic retreatment techniques.” Please, specify between which techniques you’ve found statistically significant differences.

Discussion

The sentence: “The results obtained in the present study showed the combined root canal retreatment technique 275 between Gates-Glidden drills and Hedstrom files remained a significantly higher proportion of 276 remaining carrier-based root canal filling material from straight root canal systems than ProTaper 277 Retreatment endodontic rotary instruments (p=0.018), despite employing a significantly higher 278 (p˂0.001) non-surgical endodontic retreatment working time (7.13±0.87min), than ProTaper 279 Retreatment endodontic rotary instruments (5.27±0.72min) and Reciproc Blue endodontic 280 reciprocating instrument (2.93±0.61min).” is too long. Please divide it in two sentences.

Please put a “,” between H-GG (11.56±3.72mm3) and PTR (11.47±2.45mm3) at line 282.

The sentence: “In addition, the mean 287 carrier-based root canal filling material volume of the combined root canal retreatment technique 288 between Gates-Glidden drills and Hedstrom files study group was slightly higher (12.02±3.99mm3) 289 than the mean carrier-based root canal filling material volume of the ProTaper Retreatment 290 endodontic rotary instruments study group (11.63±2.47mm3) and the mean carrier-based root canal 291 filling material volume of the Reciproc Blue endodontic reciprocating instrument (10.70±2.63mm3), 292 so there are more amount of root canal filling material to remove and it is necessary more working 293 time to achieve.” Is too long, please provide abbreviations and separate the sentence on the volume of removed guttapercha from the one on working time.

At line 321, please remove the double “the”.

The use of Micro-CT instead of CBCT and the reasons why you preferred it should be evidenced. You can use the following article to underline the fact that CBCT has been already used for this purpose in literature, but you preferred Micro-CT because of it’s advantages (please specify): “Gambarini G, Piasecki L, Schianchi G, et al. In vitro evaluation of carrier based obturation technique: a CBCT study. Ann Stomatol (Roma). 2016;7(1-2):11‐15.”

Line 343: please correct “ther higher morking time”.

References should be formatted according to instruction for authors. Please check and correct commas, semicolons and spaces.

For example: Siqueira, J.F Jr.; Rôças, I.N. Polymerase chain reaction–based analysis of microorganisms associated with failed endodontic treatment. Oral Surg Oral Med Oral Pathol Oral Radiol Endod, 2014, 97, 85-94.

Author Response

Dear reviewer 3,

I’m pleased to resubmit the manuscript of the work entitled, “The Efficacy of Rotary, Reciprocating, and Combined Non-surgical Endodontic Retreatment Techniques in Removing a Carrier-Based Root Canal Filling Material from Straight Root Canal Systems: A Micro-Computed Tomography Analysis”.

Reviewer 3: English language and style are fine/minor spell check required.

Response: In order to adapt to the reviewer 3 comment, we have sent the manuscript to the English editing service of MDPI editorial (We attached the English Editing Certificate).

Reviewer 3: Abstract: The following sentence is too long and it should be reformulated due to make the paragraph more clear: I suggest: “All teeth were submitted twice to a micro- computed tomography (micro-CT) scan before and after non-surgical endodontic retreatment procedures. Volumetric analysis of the root canal filling material (mm3), volumetric analysis of the remaining root canal filling material (mm3), non-surgical endodontic retreatment working time (min), proportion of remaining root canal filling material (%) and the efficacy of root canal filling material removing between the non-surgical endodontic retreatment techniques were analyzed using ANOVA one-way statistical analysis.”

Response: In order to respond to the reviewer 3 comments, we clarify that we have changed this sentence.-

Reviewer 3: Introduction: Line 51: [5,6,7,8] should be written [5-8].

Response: In order to respond to the reviewer 3 comments, we clarify that we have changed the reference.

Reviewer 3: Material and methods: Line 117: the sentence “…by introducing a size 25 GuttaCore obturator in the ThermaPrep heater obturator oven (Dentsply Maillefer, Ballaigues, Switzerland) to the working length.” Seems to be mispelled. Maybe the authors wanted to say that they have introduced the instrument before in the oven, then in the tooth to the working length. Please verify.

Response: In order to respond to the reviewer 3 comments, we clarify that we have changed the reference.

Reviewer 3: Material and methods: Once you’ve cited the manufacturer’s details of a product, it’s not necessary to repeat it after everytime you write the name of an instrument or whatever. Please, remove them all, instead of the first one.

Response: In order to respond to the reviewer 3 comments, we clarify that we have removed all the manufacturer’s details of a product, except for the first one.

Reviewer 3: Material and methods: Once you’ve citated the complete name of a product, it’s not necessary to re-write down it completely everytime you cite it: when and if it is possible, please use abbreviations like PTR, H-GG, etc.

Response: In order to respond to the reviewer 3 comments, we clarify that we have used abbreviations related to the study groups.

Reviewer 3: Results: Please, provide only (p<0.05) and (p>0.05) instead of (p=x.xxx), when reporting statistical significance.

Response: In order to respond to the reviewer 3 comments, we clarify that we have changed the statistical significance.

Reviewer 3: Results: Line 268: “Statistically significant differences were 268 observed (p=0.009) between the carrier-based root canal filling material removal efficacy of the non-269 surgical endodontic retreatment techniques.” Please, specify between which techniques you’ve found statistically significant differences.

Response: In order to respond to the reviewer 3 comments, we have specified the techniques that showed statistically significant differences attending to the reviewer 3 comments.

Reviewer 3: Discussion: The sentence: “The results obtained in the present study showed the combined root canal retreatment technique 275 between Gates-Glidden drills and Hedstrom files remained a significantly higher proportion of 276 remaining carrier-based root canal filling material from straight root canal systems than ProTaper 277 Retreatment endodontic rotary instruments (p=0.018), despite employing a significantly higher 278 (p˂0.001) non-surgical endodontic retreatment working time (7.13±0.87min), than ProTaper 279 Retreatment endodontic rotary instruments (5.27±0.72min) and Reciproc Blue endodontic 280 reciprocating instrument (2.93±0.61min).” is too long. Please divide it in two sentences.

Response: In order to respond to the reviewer 3 comments, we clarify that we have divided it in two sentences.

Reviewer 3: Discussion: Please put a “,” between H-GG (11.56±3.72mm3) and PTR (11.47±2.45mm3) at line 282.

Response: In order to respond to the reviewer 3 comments, we clarify that we have added the point

Reviewer 3: Discussion: The sentence: “In addition, the mean 287 carrier-based root canal filling material volume of the combined root canal retreatment technique 288 between Gates-Glidden drills and Hedstrom files study group was slightly higher (12.02±3.99mm3) 289 than the mean carrier-based root canal filling material volume of the ProTaper Retreatment 290 endodontic rotary instruments study group (11.63±2.47mm3) and the mean carrier-based root canal 291 filling material volume of the Reciproc Blue endodontic reciprocating instrument (10.70±2.63mm3), 292 so there are more amount of root canal filling material to remove and it is necessary more working 293 time to achieve.” Is too long, please provide abbreviations and separate the sentence on the volume of removed guttapercha from the one on working time.

Response: In order to respond to the reviewer 3 comments, we clarify that

Reviewer 3: Discussion: At line 321, please remove the double “the”.

Response: In order to respond to the reviewer 3 comments, we clarify that we have removed the double “the”.

Reviewer 3: Discussion: The use of Micro-CT instead of CBCT and the reasons why you preferred it should be evidenced. You can use the following article to underline the fact that CBCT has been already used for this purpose in literature, but you preferred Micro-CT because of it’s advantages (please specify): “Gambarini G, Piasecki L, Schianchi G, et al. In vitro evaluation of carrier based obturation technique: a CBCT study. Ann Stomatol (Roma). 2016;7(1-2):11‐15.”

Response: In order to respond to the reviewer 2 comments, we clarify that we have added the sentence and the reference.

Reviewer 3: Discussion: Line 343: please correct “ther higher morking time”.

Reviewer 3: References: References should be formatted according to instruction for authors. Please check and correct commas, semicolons and spaces. For example: Siqueira, J.F Jr.; Rôças, I.N. Polymerase chain reaction–based analysis of microorganisms associated with failed endodontic treatment. Oral Surg Oral Med Oral Pathol Oral Radiol Endod201497, 85-94.

Response: In order to respond to the reviewer 2 comments, we clarify that we have corrected the references

We take this opportunity to thank the recommendations and suggestions made by the reviewer to improve the document.

Yours sincerely,

Round 2

Reviewer 2 Report

The manuscript has improved, especially the English got much better. Still I see some spelling mistakes, for example, in the abstract, in the word removing, for me it seems, the e is not deleted in the end, but this might also just be a mistake of the appearance of the pdf. Please double check spelling mistakes.

Overall I am satisfied with the manuscript, congratulations to the authors.